# “Until I Know It’s Safe for Me”: The Role of Timing in COVID-19 Vaccine Decision-Making and Vaccine Hesitancy

**DOI:** 10.3390/vaccines9121417

**Published:** 2021-11-30

**Authors:** Eric B. Kennedy, Jean-François Daoust, Jenna Vikse, Vivian Nelson

**Affiliations:** 1Disaster and Emergency Management, School of Administrative Studies, York University, Toronto, ON M3J 1P3, Canada; 2Politics & International Relations, School of Social and Political Science, University of Edinburgh, Edinburgh EH8 9YL, Scotland, UK; jf.daoust@ed.ac.uk; 3Discourse, Science, Publics Lab, Department of Psychology, University of Guelph, Guelph, ON N1G 2W1, Canada; jvikse@uoguelph.ca (J.V.); nelsonv@uoguelph.ca (V.N.)

**Keywords:** vaccine, COVID-19, vaccine hesitancy, risk aversion, timing, public health

## Abstract

Managing the COVID-19 pandemic—and other communicable diseases—involves broad societal uptake of vaccines. As has been demonstrated, however, vaccine uptake is often uneven and incomplete across populations. This is a substantial challenge that must be addressed by public health efforts. To this point, significant research has focused on demographic and attitudinal correlates with vaccine hesitancy to understand uptake patterns. In this study, however, we advance understandings of individual decision-making processes involved in vaccine uptake through a mixed-methods investigation of the role of *timing* in COVID-19 vaccine choices. In the first step, a survey experiment, we find the timing of vaccine rollout (i.e., when a vaccine becomes available to the respondent) has a significant impact on public decision-making. Not only is there a higher level of acceptance when the vaccine becomes available at a later time, but delayed availability is correlated with both lower levels of ‘desire to wait’ and ‘total rejection’ of the vaccine. In a second step, we explore associated qualitative data, finding that temporal expressions (i.e., professing a desire to wait) can serve as a proxy for underlying non-temporal rationales, like concerns around safety, efficacy, personal situations, or altruism. By identifying these patterns, as well as the complexities of underlying factors, through a mixed-methods investigation, we can inform better vaccine-related policy and public messaging, as well as enhance our understanding of how individuals make decisions about vaccines in the context of COVID-19.

## 1. Introduction

While there are different ways to mitigate the spread and impacts of COVID-19 using non-pharmaceutical interventions (e.g., a wide array of preventive measures including lockdowns, physical and social distancing, etc.), vaccination campaigns around the world are essential to recovery. Their successful deployment is, however, a challenging operation. Development, approval by national health agencies, and distribution present important difficulties including logistical and ethical challenges [1,2,3]. Even when these technological and institutional factors are overcome, however, addressing individual and collective decision-making regarding vaccination remains essential—even if not sufficient on its own—for achieving public health outcomes. Successful vaccination campaigns rely on citizens’ willingness to get the vaccine, willingness to receive follow-up and booster doses, and willingness to support broad-based vaccine measures (e.g., funding for vaccine development and distribution; the emergence of pro-vaccine social norms, mandates, and expectations; viewing vaccines as a non-partisan rather than polarized issue, etc.). Coupled with other crucial factors, like availability, accessibility, and cost, this willingness is critical for a successful vaccination campaign.

Given the important role of vaccines in ameliorating the COVID-19 pandemic, tremendous attention is currently being given to documenting and improving levels of vaccine uptake. Much of this work focuses on reducing “vaccine hesitancy”, understood broadly as a “delay in acceptance or refusal of vaccination despite availability of vaccination services” [4]. While some research focuses on providing thoroughly described proportions of planned or actual vaccine uptake [5,6,7,8], a significant volume of research on vaccines—both predating and during COVID-19—has focused on identifying the factors associated with hesitance. For example, sociodemographic factors are well studied, and we know that gender, age, and ethnicity (among others) are all linked to one’s proclivity to accept a COVID-19 vaccine—as discussed in the previous meta-analyses [9,10]. Likewise, the role of individual and collective risk perceptions has been investigated in the context of the COVID-19 pandemic [11,12,13]. Other research has sought to appraise the effectiveness of educational and communicative strategies at improving vaccine knowledge and increasing willingness to vaccinate amongst populations [14,15,16].

While these demographic features, risk perceptions, and interventions via media and other campaigns can help to explain broad patterns of uptake, work remains to be done in terms of understanding the complexity of individual-level decision-making on vaccines and the unique features of the COVID-19 vaccine rollout that might affect those decisions. For example, COVID-19 vaccine decisions are complicated by a context where those living in many locations did not—or even still do not—have immediate or readily available access to COVID-19 vaccines and must think about their vaccine decisions in terms of future possibilities and uncertainties regarding access. In many jurisdictions, for example, vaccine access was prioritized by vulnerability to the disease, exposure via healthcare or other frontline roles, or other factors. Moreover, different countries have had widely variable access to dose availability. In other words, when a vaccine becomes available to an individual may vary widely and has the potential to be a causal factor in their decisions about whether or not to get vaccinated.

In this paper, we investigate the way that public acceptance of vaccines (i.e., intention to get vaccinated) varies based on scenarios of personal vaccine availability (i.e., whether the vaccine is available to a given individual immediately or at some future date). We used a mixed-methods approach that blends analysis of a quantitative experiment with a qualitative investigation of respondents’ open-ended perspectives. In the first, quantitative step, we experimentally manipulated the scenario facing respondents to isolate our variable of interest: the proximity of vaccine availability for the individual (i.e., immediately or at various points in the future) and its impact on vaccine intentions. In addition to increasing overall pro-vaccine intentions, we find that delaying vaccine availability not only reduces the proportion of respondents wishing to wait before getting vaccinated, but also the proportion of respondents who outright reject vaccination in more proximate scenarios.

In the second, qualitative step, we asked those participants who wished to wait for the vaccine to explain their reasons, which allows in-depth analysis of these perspectives. Alongside a wide variety of reasons for delaying vaccination, we find that often expressions of a desire to wait for a period of *time* are actually proxy expressions for a desire to wait for *something in particular* (e.g., certain kinds of testing, opportunities to consult with a physician, etc.). 

## 2. Reviewing the Literature: The Complexity of Vaccine Decision-Making 

Even prior to the COVID-19 pandemic, significant research attention was given to the issue of ‘vaccine hesitancy’ or ‘vaccine decision-making’ from various disciplines. Within public health and psychological research, significant work has highlighted the different roles of sociodemographic, attitudinal, and knowledge-based factors in vaccine decisions in general [17,18,19], as well as in the context of particular vaccines (e.g., influenza [20]) or communities (e.g., parents [21,22]). At the same time, critical health researchers have also problematized traditional framings within vaccine ‘acceptance’ and ‘hesitancy’ literatures, emphasizing the role of previously underappreciated factors like trust [23,24]. The example of trust is particularly illustrative of the importance of considering attitudinal and experiential variables (rather than, say, only demographic correlates), as its effect on the uptake of COVID-19 preventive measures can be larger within marginalized and/or non-dominant groups [25,26,27].

As might be expected, this massive body of literature has revealed a wide variety of factors that affect vaccine decision-making. Beyond the sociodemographic, attitudinal factors like beliefs, risk perceptions, and the behaviors of oneself and others play a critical and interconnected role in vaccine decision-making [4,5,28,29]. More specifically, elements like prior vaccine behavior [30], attitude toward vaccines more broadly [31], and impacts of compulsory or mandatory rules around some vaccines [32] can play a role in influencing decisions. In the context of COVID-19, decision-making has been connected to political affiliation [12], ideological and partisan factors [33,34], (mis/dis)information [35,36], and appraisals of the quality of other government decision-making on COVID-19 issues [37], to name just a few. In short, the question of vaccine uptake involves a large number of attitudinal factors; a complex ecosystem that has been described through various models, including socioecological frameworks [38].

The influence of these factors—aspects like political affiliation, partisanship, information quality, and trust in government decisions—reveals a landscape wherein individual perceptions of risk, safety, and efficacy are critical to vaccine decision-making. This is supported by earlier research (e.g., efficacy [39,40]; safety [41,42]; risk perceptions [43,44]), as well as more recent experiments exploring the role of different kinds of vaccine messages on stated vaccine intentions [45]. Moreover, this is reinforced by in-depth qualitative work [46], wherein respondents self-identify safety and efficacy as critical issues in their decision-making about vaccines. 

Of course, vaccine decisions are also more complex than simple binaries of ‘taking’ or ‘not taking’ a vaccine. For example, scholars [23,24] have argued that traditional framings of vaccine ‘hesitancy’ rely on problematic ‘deficit models’ and simplify the many complex processes involved in making a decision to either take *or* not take a vaccine. Adopting a more ‘symmetric’ approach to studying vaccine decision-making of both those who chose to receive and not receive vaccines (as opposed to simply focusing on the hesitant), as well as appreciating a spectrum of different kinds of concern (from outright rejection to the desire to ‘wait and see’) can offer a more holistic perspective that avoids assuming ‘deficits’ amongst those with concerns (for the broad principle of symmetry in analysis, see [47]).

In the case of COVID-19, this is made even more complicated by concerns among members of the public that the vaccine may have been ‘rushed’, which has been documented to impact vaccine intentions and vaccine trust [48,49]. This has led to the emergence of a common ‘wait and see’ position, wherein some who plan to eventually get a vaccine report an intention to wait for a period of time before doing so. For example, one study [37] documented a majority (56%) of respondents in Portugal wishing to “wait”, a decision that was associated with being younger, losing income during the pandemic, and having less confidence in government and public health responses, among other factors. Likewise, in Japan, “wait-and-see” represented a significant proportion of survey participants, with those respondents citing side effects, altruism, and personal vaccination being unnecessary if enough others get vaccinated as the most common reasons for delaying [50]. Some existing work [39] provides reason to believe these issues of timing might indeed be related to perceptions of ‘rushed’ development, as experimental evidence suggests that vaccines that took longer to develop are more preferred by American respondents.

In the Canadian context, there has been a comparatively high degree of professed vaccine acceptance—and actual vaccine uptake. Public opinion polling firms initially showed just under 80% of respondents intending to get vaccinated, with increases to near 90% of respondents affirming that intention by May 2021 [51]. Even this early broad support, however, showed significant qualification: in September 2020, only 39% of respondents intended to get vaccinated ‘as soon as possible’, while 38% intended to get vaccinated but wished to wait before doing so. This large proportion of people stating a desire to ‘wait’ raises a number of interesting questions worth considering, both in the context of COVID-19 and vaccine uptake more generally: Why did such a large group state a plan to wait—or are perhaps still waiting, even to this day? What factors influenced (or are influencing) their decision-making about how long to wait? Moreover, how closely did this stated intention to wait, line up with actual plans, as opposed to idiosyncrasies of that particular formulation of the question? 

To address these questions—and to advance the understanding of determinants of waiting behaviors—we integrated a series of questions into a mixed-methods, nationally representative survey of Canadians. We first investigate the impact of different timing scenarios (e.g., at what point the vaccine becomes available to the respondent) using an experimental design, and we then explore the qualitative explanations provided about these decisions. We detail our mixed-methods approach in the next section.

## 3. Methods

To investigate the influence of vaccine availability on vaccine intentions, we designed a scenario-based survey experiment to isolate the issue of personal availability of the vaccine, followed by a qualitative investigation into open-ended rationales offered by participants. This experiment was administered as part of a national Canadian survey on COVID-19 risk perceptions, impacts, adaptations, and preferences [52]. All participants completed an informed consent process and were allowed to opt out at any point.

In this paper, we consider a subset of 2602 respondents recruited from the Leger polling firm online panel between 9–22 December 2020, a timing selected to be just prior to broad vaccine availability, but while there was already a great deal of anticipation, information, and public discourse surrounding the eventual rollout. These participants were recruited using nested quotas factoring in provincial populations, gender, age, and visible minority status. All recruitment and responses occurred during the aforementioned period, in multiple waves to ensure completion of each quota category. Once quota was met, no further respondents were recruited within those demographic criteria. Overall, the sample closely mimics the Census data on variables like age, gender, provinces (see Appendix A for more detailed comparison). The survey also collected responses on a variety of other issues, including demographic variables which were investigated as possible cofounders in a further robustness check later in the results section.

In a section of the survey asking about vaccine intentions, respondents were presented with a simple scenario: “If a coronavirus vaccine was available to you [today/in one month/in six months/in one year], would you get vaccinated, or not?” (To see how the user would experience these questions, see the screenshots in the Appendix A.) Using a ‘soft require’ format (in which participants were reminded if an answer was missing, but allowed to proceed if they clicked ‘next’ again), participants then had the option of choosing from six options: Yes, I would get a vaccination as soon as one became available to meYes, I would get a vaccination, but would wait until some time passes firstYes, I would get a vaccination, but would wait for something elseNo, I would not get a coronavirus vaccinationPrefer not to sayUnsure

Options 1, 2, and 4 were drawn from the ongoing Angus Reid COVID-19 tracker [51] which provides frequent national data in the Canadian context on vaccine attitude and allowed us to use a measure that has already been deployed in a broader context. Option 3 was added with the aim of differentiating between temporal and other rationales behind variations, and issues identified in a preceding round of qualitative interviews as part of the larger COVID-19 project [53]. 

Participants were randomly assigned to each of these four experimental variations using the randomization within the survey platform (Qualia Analytics). Each respondent had a probability of 0.25 to receive any of the four treatments. Randomization is key as it allows us to be very confident that all potential cofounders are equally distributed across the different groups. The use of a between-subjects experimental design was essential in order the estimate the average causal effect of different timelines of the vaccine offers. For instance, we would not expect that participants could, necessarily, explicitly articulate the factors affecting a complex decision; nor would we be sure that the factors they did articulate were actually those that would drive the behavior. If participants had been presented with all four scenarios (i.e., a within-subject design), for example, participants might have felt pressure to maintain consistency across answers. For instance, if a participant selected “wait until some time passes” during an early scenario, they might be more likely to acquiesce during a later scenario. Similarly, if they selected “no, I would not” for an early question, they might be tempted to answer the same way for all future questions (a phenomenon we found did not hold true; see Section 4.1). By foregrounding the issue of vaccine intentions—and concealing the temporal variation by only presenting participants with a single scenario—our aim was to improve the quality of data collected.

To enable the second portion of the investigation—the qualitative analysis—if respondents selected option 2 or 3 (that is, wait for a period of time or wait for something else), they were then presented with a follow-up question and open-ended text box asking them to describe “how long they might wait” or “what they planned to wait for”, respectively.

As is the case with many survey questions, these questions are subject to the possibility of social desirability bias, wherein respondents might believe that there is a social norm in favor of vaccination. While the literature is mixed on the presence of a social desirability bias in the COVID-19 literature [54,55], the main issue relates to a potential heterogeneous effect of social desirability bias across subgroups. Fortunately, recent research [56] has shown that this bias seems to be homogenous across respondents. Hence, even if there was a bias boosting the positive vaccination intentions, previous literature suggests that this should hold consistent across respondents to all variations of the question. 

Once the responses were gathered, data cleaning involved three verification questions. First, participants were asked to provide their age range as well as the year of birth at different points in the survey. Instances where answers were in conflict were identified. Second, participants were asked to provide both their province of residence and their postal code (which was used to deduce province of residence). Respondents who provided contradictory information were identified. Third, an explicit attention-checking question was embedded in the survey. All participants who were identified in these three steps were manually checked for having provided cogent text responses to at least one open-ended question. Those who did not were removed from the dataset. 

Following data cleaning of all responses, we turned our attention to the qualitative data. Some *n* = 892 respondents provided answers to these open-ended questions, which were then coded inductively by three members of the research team, allowing multiple codes to be applied per piece of data. After developing an inductive codebook based on an initial review of the data, we used an intercoder consensus approach (which is distinguished from traditional intercoder reliability approaches [57]) to allow team members to discuss places of disagreement in real-time and to ensure that the corpus was exhaustively examined by all coders. Because of the relatively straightforward nature of the short text-based responses, disagreements were rare and generally encompassed cases of either ‘other’ or places where multiple codes applied.

## 4. Results

### 4.1. Statistical Analysis of the Survey Experiment

Given the clean design of our survey experiment in isolating the variable of interest, we directly focus on the bivariate relationship, that is, vaccine intention across the treatments. It also allows us to include every respondent in our sample (as they are not missing on other variables that we would include), with the only exception of 149 participants that we excluded due to the fact that they were not Canadians. The four groups were composed of 622, 661, 670 and 615 respondents, respectively. 

In a first step, we conducted a Pearson’s Chi-square test (using Stata *chi2* option) when cross-tabulating vaccine intention across the four treatments. As expected, we find a difference that is statistically significant, whereas X^2^(9, N = 2568) = 27.466, *p* = 0.001. In a second, we unpacked the differences among the answer choices and the treatments. To do so, we conducted multinomial logistic regression shown in Table 1 below. To better represent the differences, we estimated the predicted probabilities (using the ‘margin’ command in Stata 16) based on the regression model of Table 1 and visualized the findings in Figure 1 below. We include 95% and 84% confidence intervals [58]. As demonstrated in Figure 1, the proportion of respondents who choose the “Yes, as soon as possible” category, that is, the most enthusiast respondents, substantially increase as the scenario of vaccine availability becomes more temporally distant. In the most proximate scenario (“today”), about 44% reported a desire to get vaccinated as soon as possible. This percentage increases to approximately 46% (“In one month”), 50% (“In six months”), and 55% (“In one year”). 

The effect is very much linear, although we should note that the biggest impact comes from the six-month treatment (compared to any of the other conditions). Overall, the total effect (from “today” to “in one year”) is about 10 percentage points, which is very substantial and statistically significant at *p* < 0.05 (as shown by the confidence intervals), especially when we consider the implications of vaccination on the number of hospitalized citizens and deaths [59] and the need for broad vaccine uptake to reduce transmissibility. A more conservative comparison, for example, contrasting the proportion of those vaccine-enthusiast respondents between the “today” and the “in six months” treatment, still lead to a large effect size with an increase of 6 percentage points. 

The percentage increase in those intending to receive the vaccine as soon as possible across treatments is accompanied by a decrease in other answer choices. Importantly, this decrease comes from *both* those who mention that they would take the vaccine but would wait, as well as from those who would have otherwise stated an intention not to take the vaccine at all. In both cases (i.e., from the “wait” and “no” groups), the decrease is about 7%, which is especially important when comparing the proportion of 18% who said “no” to 11% of these across the treatments (from “today” to “in one year”). We should note that the “yes, but would wait for something else” category is remarkably stable and always included less than 5% of the sample. 

While we believe that Figure 1 presents the clearest depiction of the experiment, one might think of other ways to look at the findings. First, the inclusion of the “unsure” might be interesting as it could encapsulate a significant form of hesitancy (e.g., uncertainty leading to caution). Appendix A shows that including these respondents (about 5.5% of the sample) does not alter our main findings. Second, we believe, following existing work [60], that including covariates in order to estimate the treatment effect should be done cautiously, but we should note that controlling for age, gender, education, province and francophonicity does not alter our findings (see Appendix A). This also allows us to be confident that the randomization between the four treatment groups did not introduce a systematic bias with respect to factors like gender, age, region, etc. Controlling for regions is particularly reassuring as Canada is a vast country with quite a lot of regional variation on different outcomes. While vaccination is the most important preventive measure, scholars have examined whether there were some regional differences in several preventive measures (including social distancing, wearing a mask, etc.) across the country. Quite surprisingly, previous work [56] involving 23 survey waves with more than 22,000 respondents over twelve months found that regional differences, when any, were modest. Moreover, we can also perform similar checks (see Appendix A) to verify that other important factors—like the perception of COVID risk or risk of vaccines—were also equally distributed across the treatment conditions which, indeed, they were.

Overall, there is strong evidence that timing affects citizens’ willingness to take the vaccine: the longer until a vaccine becomes available, the more respondents become willing to take it immediately, and the fewer state a desire to wait—or to not take the vaccine at all. This research, therefore, illustrates an additional factor to add to the many already within the literature reviewed in Section 2—that is, time until personal availability of a novel vaccine. This result, however, opens additional questions: what factors do participants use to explain their desire to delay receiving the vaccine?

### 4.2. Qualitative Analysis of the Open-Ended Responses 

Building upon the quantitative experimental design of the vignettes, our research design also allowed us to dive into more fine-grained analyses of respondents’ vaccine intention using open-ended questions. Open-ended questions varied across answer choices. Respondents who selected “Yes, but would wait until some time passes first” were presented with the question “How long do you think you might wait?” For respondents that selected “Yes, but would wait for something else first”, they were asked, “What do you plan to wait for?” In total, some *n* = 892 qualitative responses were given to these questions, ranging from a few words (e.g., “six months” or “evidence of safety”) to more detailed responses (e.g., “Until I’ve received some reassurance that the vaccination is more on the safe side for me to take with my current health issues”).

#### 4.2.1. Inductive Coding: An Overview of Responses 

Using the inductive coding scheme with consensus-based coding described above, researchers sorted the responses into a series of codes that fit within three broad categories: temporal, substantive, and other. Temporal codes identified a specific period of time, either from a milestone within vaccine development (e.g., waiting until X months after federal approval) or from point of availability (e.g., waiting until X months after it “becomes available to me”) For future researchers working on this topic, we highly recommend careful design of these temporal questions. In particular, because of the brevity of many of the qualitative responses (e.g., “six months”), it was very difficult to assess whether respondents were answering the question based on their date of response (e.g., “six months from today”) or based on their scenario’s timing (e.g., “six months from the scenario I was just presented with, which is the vaccine becoming available to me six months from today”). This lack of confidence that all respondents interpreted the prompt the same way led to the inability to confidently analyze the variation in temporal responses, which is why we treat them as a single category within this article.

Substantive codes were those that identified specific issues or concerns, which fell into the following clusters:Safety (e.g., see if it causes side effects)Efficacy (e.g., see if it actually reduces symptoms or transmission)Ambiguous safety/efficacy (e.g., references to ‘needing more evidence’ or ‘needing to see more’, but not explicitly identifying whether they were looking for either safety or efficacy)Health conditions (e.g., suitability given personal situations like pregnancy, allergies, etc.)Priority (e.g., responses including either altruistic ‘let those at higher risk get it first’ or realistic ‘it will be hard to book until’ responses)Purpose (e.g., need to get for travel, work)Leaders (e.g., want to wait until politicians, doctors, or celebrities get it themselves)Other (a reason that did not fit into the above)Unsure (e.g., explicitly said they were not sure what or how long they wanted to wait for)

Finally, some responses (n = 120) were deemed to be unable to be coded, either because of insufficient information to make any judgment about the subject matter or because of irrelevance to the question asked. Note that each response could contain more than one code (e.g., several respondents explicitly identified both ‘safety’ and ‘efficacy’).

Among the coded responses, there was a very clear pattern: there was a dramatic difference between respondents who said they were waiting for some time to pass versus those who said they were waiting for something else (see Figure 2). For those who were waiting for some time to pass, approximately 3 in 5 respondents (59%) indeed specified a specific timetable ranging from weeks to years before they would feel comfortable getting the vaccine—as contrasted with only 5% of respondents who said they were “waiting for something else” but then went on to specify a time period. Yet, among those respondents who said they were waiting “until some time passes first” before getting the vaccine, almost 1 in 5 (18%) articulated not a temporal but substantive rationale, while another 9% expressed that they were unsure. By contrast, almost three-quarters (73%) of respondents who said they were “waiting for something else” expressed substantive criteria, with another 22% of their responses uncodable—and, notably, not a single expression of being unsure what they were waiting for. 

Perhaps the more striking difference, however, comes from comparing the substantive and uncertain rationales between these two groups of respondents (see Figure 3). For respondents who indicated that they wished to wait “until some time passes first”, the dominant explanation was that of uncertainty: respondents explicitly expressed uncertainty when asked the follow-up question “how long do you plan to wait?” The next most common responses involved safety concerns (28%), questions of priority (i.e., feeling as though others deserved to get the vaccine first; 10%), and efficacy concerns (9%).

By contrast, not a single respondent who said they were waiting not “until some time passes first” but “for something else” expressed uncertainty about what they were waiting for. Rather, this group overwhelmingly expressed concerns about safety (41%), ambiguous forms of safety and/or efficacy (see below for more discussion; 25%), and efficacy (15%). Again, these were not closed-ended prompts that primed the respondents (nor were any of these factors introduced in the survey prior to this question): these were open-ended expressions by the respondents about what they believed would move them from the category of ‘waiting’ into actually obtaining the vaccine.

#### 4.2.2. Close Qualitative Reading: Exploring the Articulations of These Codes 

In addition to coding these responses into the themes discussed above, we were also able to explore the open-ended data for more nuanced expressions of individual attitudes, beliefs, and opinions. This allows for richer insights into the motivations driving these behaviors, as well as an opportunity to understand the heterogeneity of concerns present among those who self-identified as wishing to wait to receive the COVID vaccine. It also provides the groundwork for future qualitative inquiry into the perspectives of those who choose to delay their vaccination.

As indicated in the coding, the single most common specific answer (excluding uncertainty) was to raise concerns about the safety of the vaccine. This, unsurprisingly, includes significant discussion about ‘side effects’, which cut across a wide range of time horizons and cases. For example, some respondents seemed to be concerned with side effects at the time of injection (e.g., “I’d wait until vaccinations had been going on for a little bit to hear if there were any common serious side effects”, stated one respondent), while others articulated much more explicitly long-term time horizons (e.g., “six to eight months maybe… depends on the reactions people are having with the vaccine”). These concerns with side effects were often explicitly connected to perceptions of rushed development, such as one respondent who stated “Side effects. It’s too early to say that a vaccine is ready for use and 99.9% safe”, or another respondent who said, “I think it takes about 2 years for a vaccination to be made and tested as safe”. This perception of rush leading to a lack of safety was, at times, connected explicitly to perceptions of the trustworthiness of the pharmaceutical industry, such as by the respondent who said, “The vaccine was developed and marketed quickly, there are no data on efficacy and long-term side effects yet”.

These concerns about safety were often cast in an explicitly epistemic nature: individuals grappling with how they could personally know and trust that the vaccine was safe. For instance, some respondents referenced the need to see *others* get the vaccine first, such as the statement “To see what side effects it has on others who want to be the Guinea pigs”. Sometimes this was delivered in a hyper-local fashion, like a respondent who wanted to wait “A couple months… Until other people in my province have had it and there seem to be no side effects” rather than relying on trials from elsewhere. Still, others called out specific kinds of knowledge that they thought were lacking, such as long-term trials (e.g., “Once 1 year study data is available from the clinical studies”) or from more specific subpopulations (e.g., “to make sure it is really safe for me and people like me”, which was coded as both safety and health conditions).

By contrast, other respondents articulated concerns with efficacy: uncertainty about whether the vaccine was actually as efficacious as claimed. Often, this was articulated as a divergence between ‘lab’ and ‘real-world’ demonstrations of effectiveness, such as waiting “long enough to see that it works” or “until data is available proving the vaccine works well” became available.

Relatedly, some responses referred to more ambiguous concerns about the safety and/or efficacy of a COVID-19 vaccine, without explicitly mentioning “safety” or “efficacy”. These responses were coded as “ambiguous safety/efficacy”. Some responses made mention of waiting until the vaccine could be deemed trustworthy (e.g., “until it can be trusted it’s still new” or until “more people get tested”), whereas other responses referred to waiting to observe vaccine “effects” without specifying which effects were of interest (e.g., “until the effects of having been vaccinated can be visibly demonstrated and observed” or “depends on how the response will be after some people get the vaccine”).

Some of the most interesting and nuanced answers arose from those who respondent by citing concerns about individual health conditions, such as referencing personal ailments or physiological characteristics that might influence their decisions. For example, one respondent cited waiting “Until I’ve received some reassurance that the vaccination is more on the safe side for me to take with my current health issues”, while another suggested holding off “until I talk it through with my doctor to make sure I do not have a medical reason that I should not take it”. Pregnancy—specifically, concerns about a lack of thorough trials, or sufficient safety data—was an oft-cited personal situation here, including respondents who gave responses like “Until after I give birth since I am expecting and I am unsure if they did a clinical test for pregnant people” or “When there is enough research to show its efficacy and when I am no longer breastfeeding, as there is not enough research data on the vaccine’s impact on breastfed infants”. Likewise, allergies were another common issue mentioned, such as waiting “until info about allergies were available and a safe setting with EpiPen on hand”, as well as responses that included the desire to wait for condition-specific data (e.g., “until I see research data that says it’s pretty safe for me, as I don’t think I’ve ever had an RNA vaccine before and I do have some allergies to molds (+Penicillin) and nuts and leaves”). 

For some participants, the decision to wait was not about any form of hesitation, but rather either a ‘reality-check’ that they were unlikely to get access soon because of government priorities or pro-social altruism, believing that they owed it to others to allow those at higher risk to get their shots first. An example of a ‘reality-check’ response would be the participant who articulated planning to wait “due to my age, I would have to wait a few months before it is our turn to get the vaccine. The priority is frontline medical workers and seniors”). By contrast, an example of pro-social altruism would be the respondent who suggested that they would wait “Not very long, but I am not a high-risk group so while supplies are limited I would let the doses go to others first”. 

A small number of respondents mentioned waiting until vaccination was necessary for some other area of life (e.g., “Until I need to travel”, or “Until I am back at work”). Such responses were coded as “purpose”. Only three responses made any sort of mention of waiting until leaders, like politicians (“to see if the damn politicians get it first”), social influencers, or cultural leaders. These responses were coded as “leaders”.

Some responses included meaningful information about how long participants might wait to vaccinate, although the reasons provided did not fit into one of the aforementioned themes and were coded as “other”. This included a wide range of one-off answers, such as crowds (“Until the big rush is over. I remember the crowds and the long line-ups for the H1N1 vaccine and would not want to repeat that experience.”), general knowledge (e.g., “Depends on when I know more details about the vaccine” or “Until I’m knowledgeable about it and comfortable getting it.”), or more ambiguous forces (e.g., “until I get the urge to go”).

Finally, some respondents made explicit mention of not knowing what or how long they wanted to wait for (e.g., “that’s so hard to say. I desperately want to see my family, but I don’t want possible health problems from the vaccine. I feel like I’m damned if I do and damned if I don’t” or “not sure at this point”). Such responses were coded as “unsure”.

## 5. Discussion

In this investigation, we used a mixed-methods design intended to assess the impact of vaccine availability and timing on individual decision-making. Through the design of the study, we were able not only to isolate the effect of the timing element but also to investigate the rationales that these individuals give for wanting to delay their vaccinations. In doing so, this work contributes to the existing literature by providing a clearer understanding of the factors that go into vaccine decision-making (beyond the demographic and ideological correlates), and also presents a rich and nuanced portrait of some of the factors being considered that can often be lumped under the heading of ‘hesitancy’.

Through the experimental design, we found that there was a strong and influential effect from the timing component. The more temporally distant the availability of COVID vaccines, the more likely individuals were to want to get the shot immediately. Perhaps most saliently, however, this increase in vaccine uptake does not just come from converting those who would state a desire to “wait” in a more proximate scenario. Rather, it results from *people who would have otherwise stated no desire to get the vaccine becoming more open to receiving it*.

This has important implications for policymakers. As the COVID vaccine rollout is underway around the world, it is important to remember that differing individual preferences can be a ‘feature rather than bug’. In other words, the fact that some individuals want to be first movers, while others want to delay their vaccination slightly, can actually serve as an advantage in rolling out a supply-constrained vaccination campaign. Moreover, our evidence shows that these late adopters may actually not be an impossible group to reach while they express hesitations, questions, and concerns, they also show a proclivity to increase in willingness to get vaccinated as time passes—even if they might otherwise have said ‘no’ in an earlier offering of the vaccine. Combing these two approaches (embracing delay through a diversity of strategies to reach early, middle, and late adopters, and dealing with these substantive concerns in empathetic, understanding, and validating ways) may be key in transitioning vaccination campaigns from initial victories to broad population coverage. 

The insight of this mixed-method design, therefore, is also useful. As we found, there are a wide variety of reasons for waiting. Some of these might be wanting time to pass, but even amongst those who identify as wanting to “wait some time”, a portion actually identify non-temporal concerns as motivating their delay. Indeed, almost no answers in the entire survey embody conspiratorial or wildly off-base concerns. Instead, respondents generally seem to raise highly salient, well-informed, specific, and appropriate questions when given the chance. This paucity of conspiratorial attitudes with respect to the vaccine question is striking, as these sorts of attitudes were well represented among respondents throughout other questions in the survey. However, it is important to note that the qualitative component of this particular study solicited only responses from those who intended to *wait* for vaccines, rather than those who rejected them entirely, where it might be expected that a significant portion of the conspiratorial sentiment may be represented. In any event, the deployment of effective communication strategies [14] that help to provide targeted information that addresses these questions may be useful. This is particularly important given the negative impact of (dis/mis)information on vaccine behavior around the world [61,62,63]. Scientific communication must take into account readers’ and viewers’ concerns, should be simple and accessible, and should ideally also be capable of performing well in a social media environment [64]. 

Another important finding of this work is the fact that nearly one-in-ten respondents who said they were waiting for some time to pass were unsure about what they were waiting for (i.e., neither a pre-determined amount of time nor for a particular issue to be addressed). This group of respondents represents an important group for further investigation—and for empathetic outreach. While some folks waiting may have specific concerns (e.g., safety) that can be addressed through listening, dialogue, and caring engagement, others—at least on the surface—seem more uncertain about their reasons for waiting and may require different engagement strategies to help work through considerations that are less articulated.

There are, of course, limitations to this work. As we are investigating in the context of COVID-19 via another portion of the overall project, the degree to which online panels truly represent broader populations is subject to significant debate. Likewise, it is critical to recognize that measures of ‘intention’ capture only professions of anticipated behavior, not the actual behavior itself (e.g., someone might plan to wait to get vaccinated for some time, but actually be swayed by incentives tomorrow). As such, we can speak to the influence of time on respondents’ anticipations of their own behavior but cannot confirm whether their real-world actions matched these stated intentions. Moreover, we use Canadian data exclusively given that, unfortunately, this original survey was not conducted anywhere else. While we believe that is it reasonable to expect a similar effect of timing on individuals’ vaccines behavior elsewhere, more data from other countries is needed to establish the external validity of our findings. It is also challenging to know whether our findings would apply to different vaccines (e.g., malaria, influenza, etc.), as the broad societal discourses around the ‘novelty’ and ‘speed of development’ of the COVID-19 vaccine reflect the unique priority and context of its development [65,66]. In other words, timing may be a comparatively important factor in decision-making in this context because of the unique COVID context. Finally, we would note that this mixed methods design suggests the importance of a great deal more qualitative *and* quantitative investigation on these topics. For instance, there are many more correlates that could be explored related to vaccine delay (e.g., mental health, different measures of risk perception, local vaccine availability, etc.). Likewise, the qualitative data calls out for further data collection in richer qualitative ways (e.g., interviewing or focus groups, rather than just a short answer survey text).

Regardless of these limitations, however, this work challenges traditional notions of vaccine hesitancy. It is easy to label those who want to ‘wait’ as simply being “vaccine hesitant”. While this may be fitting if “hesitant” is taken in its literal form, “to hesitate”, the label of “vaccine hesitancy” can pathologize and dismiss this group, reducing complex, nuanced, and insightful questions into a binary of ‘those who accept science’ and ‘those who don’t.’ Our study adds to the literature that questions this traditional framing, thereby reinforcing the need to move beyond deficit-based, strictly educational techniques of public outreach to those that take questions seriously, empathetically, and kindly, and work to provide actual and individualized answers that validate the thoughtfulness of the person making decisions about vaccination.

Finally, this work has important methodological implications as well. Studies of vaccination intentions need to abandon attempts to measure intended or actual behavior as a dichotomy. Not only do those ‘waiting’ provide an interesting middle-ground case worthy of understanding, but there are many different forms of waiting. While those who identified “waiting for something else” other than time were less common in our sample, a non-trivial portion of those who identified as waiting for a period of time actually articulated waiting for something else entirely. It is important to use methods that can capture this granularity and diversity, as appreciating these heterogeneous underlying reasons allow for more robust work to support pro-vaccine decision-making.

## 6. Conclusions

In this study, we used an experimental design in a large-scale Canadian COVID tracking study to better understand individual decision-making. We found that the timing presented to respondents (i.e., is a vaccine available to you today, in a year, or somewhere in between) has a significant and substantial effect on intended behaviors. Indeed, this study suggests there may even be, as time passes, possibilities to convert those who would have described themselves as entirely unwilling to receive a vaccine in a more proximate scenario. Moreover, we found a wide variety of reasons for ‘waiting’ among those who self-identify in that category, including a non-trivial proportion of respondents who articulate wanting to wait until time passes, but actually use that as a proxy for waiting for something else (e.g., improved data on safety, evidence of efficacy, or uncertainty about the decision as a whole). By paying attention to this variety of responses, we can better tailor outreach efforts and engagement strategies. Through this mixed-methods investigation, we enhance the understanding of vaccine decision-making and provide tools for supporting the COVID-19 (and other) vaccine roll-outs worldwide.

## Figures and Tables

**Figure 1 vaccines-09-01417-f001:**
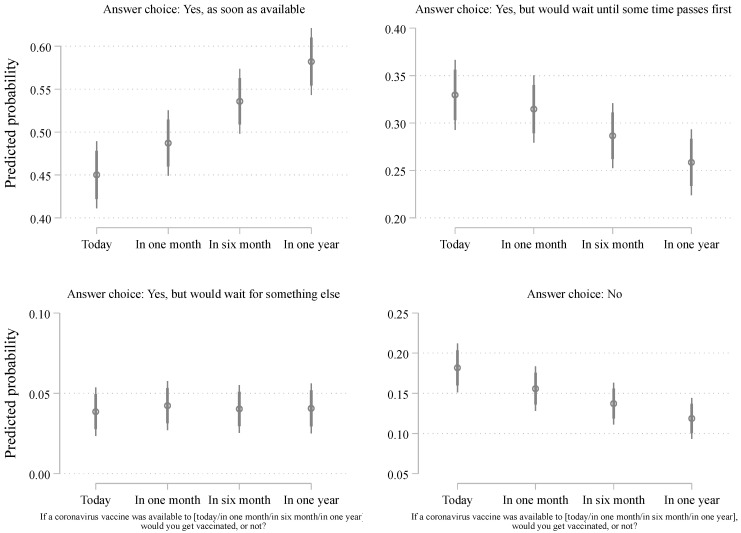
Distribution of vaccine intention across experimental conditions. Note: estimates from Appendix A. 95% and 84% confidence intervals included.

**Figure 2 vaccines-09-01417-f002:**
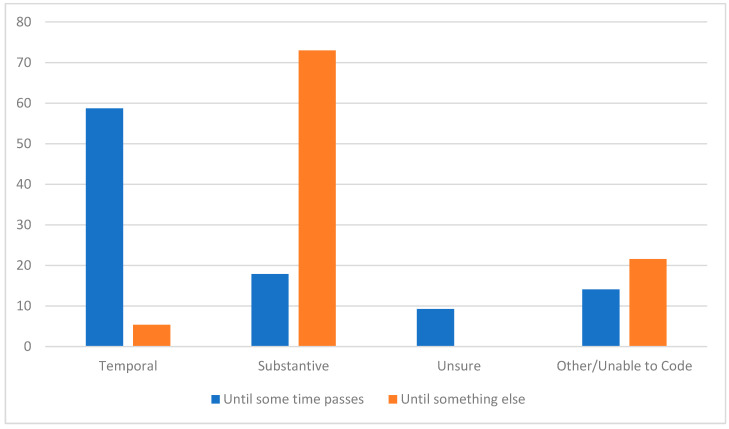
Percentages of responses coded within the categories of temporal, substantive, unsure, or other/unable to code.

**Figure 3 vaccines-09-01417-f003:**
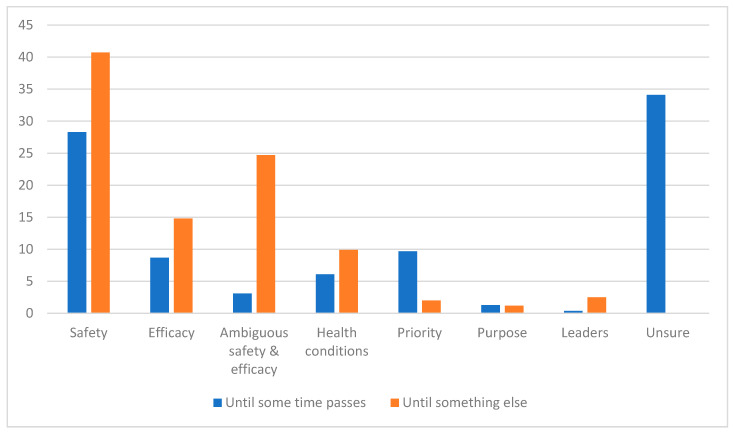
Breakdown of substantive rationales by respondent answer.

**Table 1 vaccines-09-01417-t001:** Multinomial logistic regression.

Outcome:	Yes… Until Some Time	Yes… Something…	No
In one month	0.88	1.01	0.79
	(0.11)	(0.29)	(0.13)
In six month	0.73 *	0.88	0.63 **
	(0.09)	(0.26)	(0.10)
In one year	0.60 ***	0.81	0.51 ***
	(0.08)	(0.24)	(0.08)
Constant	0.73 ***	0.09 ***	0.40 ***
	(0.07)	(0.2)	(0.04)
Observations	2568		
Pseudo *R*^2^	0.005		

Note: Odds ratio with standard errors in parentheses. The base outcome is “Yes, as soon as possible”. The reference category for the treatments is “Today”. * *p* < 0.05, ** *p* < 0.01, *** *p* < 0.001.

## Data Availability

Following the completion of the study, the de-identified full survey dataset will be available open-access, as well as replication files for this analysis; see https://www.cemppr.org/research/covid-19-in-canada for updates (accessed on 16 November 2021). This data is also available, prior to open access upload, via reasonable request to the corresponding author.

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
