# Peer review of "“Until I Know It’s Safe for Me”: The Role of Timing in COVID-19 Vaccine Decision-Making and Vaccine Hesitancy"

_vaccines, 2021, doi:10.3390/vaccines9121417_

Round 1

Reviewer 1 Report

After the first submission, the paper was improved and it is now suitable for pubblication

Author Response

We thank R1 for the positive words and their verdict.

Reviewer 2 Report

The manuscript “Until I know it’s safe for me”: The role of timing in COVID-19 vaccine decision-making and vaccine hesitancy” by Eric B. Kennedy and colleagues provides and discuss findings accordingly to the aim and scope of the journal and in particular those related to regulatory affairs, commercial utilization, policy, safety, epidemiology of vaccines.

The study investigates how the public acceptance of vaccines varies based on scenarios of personal vaccine availability and individual decision-making processes – mainly focusing on reducing vaccine hesitancy - by a quantitative and qualitative mixed methods approach, though limited to Canada.

They use a survey experiment, administering people various questions about timing of their willing to be vaccinated, and annexed motivations and found that Canadian people’s individual decision about COVID-19 vaccination can bring to a better tailored vaccine rollout strategy.

The matter in discussion is extant and obviously very interesting, and could be better related with potential impact on government decision about vaccine supplies and rollout. Nevertheless, it should be exploitable for various countries beyond Canada. In fact, as the authors suggested this study may “enhance our understanding of how individuals make decisions about vaccines in the context of COVID-19 and beyond”; however, they do not provide any other “beyond” proof of concept.

Moreover, although this study holds a pretty interesting argumentation I wonder if their findings lie more on the peculiar pandemic situation than actual and usual decision-making process of individuals in a normal scenario. In other words, do the authors think the individual decision-making process would be different in another context, as for influenza vaccine or polio or the most recent malaria? How the mass media information and infodemia (including fake news) interfere with this process? It would be interesting to discuss these factors too.

Moreover, not sure whether this approach may help in the administration decision as does not consider any geographical distribution likely resulting in an inefficient vaccine distribution hence herd immunity if targeted only to willing people that may concentrate more in a specific area than another.

This paper is very well written, nevertheless in this form it’s quite long winded, I would suggest to be more concise and straight to the point.

Finally, I would really appreciate they discuss more the limitations of the study, which would highlight a strong analytical sense and scientific responsibility, meaning that they got the weaknesses of the study and its design from different point of view.

I would suggest to rewrite it in order to get a clearer and more convincing narrative.

Author Response

In their assessment, Reviewer 2 raises five main comments to be addressed.

Comment 1: “The matter in discussion is extant and obviously very interesting, and could be better related with potential impact on government decision about vaccine supplies and rollout. Nevertheless, it should be exploitable for various countries beyond Canada. In fact, as the authors suggested this study may “enhance our understanding of how individuals make decisions about vaccines in the context of COVID-19 and beyond”; however, they do not provide any other “beyond” proof of concept.”

We thank R2 for this comment. We agree that the reference to ‘and beyond’ in the abstract is poorly justified and we have removed that wording.

Moreover, in line with the second comment of R2 (see below), we realized that we were not explicit enough about the scope of our research. We must acknowledge that we do not, as mentioned by R2, have evidence of external validity beyond Canada --- even if we agree that it is reasonable to expect our findings to replicate in a variety of contexts outside this country. The reason is simple: we make use of an original dataset that based on an exclusively Canadian survey we conducted. We now explicitly highlight this limitation in the revised version of the manuscript through an added passage on page 14:

“Moreover, we use Canadian data exclusively given that, unfortunately, this original survey was not conducted anywhere else. While we believe that is it reasonable to expect a similar effect of timing on individuals’ vaccines behaviour elsewhere, more data from other countries is needed to establish the external validity of our findings.”

Comment 2: “Moreover, although this study holds a pretty interesting argumentation I wonder if their findings lie more on the peculiar pandemic situation than actual and usual decision-making process of individuals in a normal scenario. In other words, do the authors think the individual decision-making process would be different in another context, as for influenza vaccine or polio or the most recent malaria? How the mass media information and infodemia (including fake news) interfere with this process? It would be interesting to discuss these factors too."

We thank R2 for this useful point. Indeed, there are at least two ways this factor could play out uniquely in the context of COVID-19. First, because the vaccine development looked different in this pandemic (i.e., a novel vaccine being developed for a novel threat) rather than ongoing supply of a more ‘established’ vaccine, the question of timing is foregrounded in this case in a way that it is not with other diseases. To this point, like with the previous comment, we have removed the ‘and beyond’ wording to help make clear that we’re not claiming this process would occur in the same way or to the same extent in other cases (e.g., other vaccines mentioned by R2). We have also added additional text on page 15 in the discussion to address this:

“It is also challenging to know whether our findings would apply to different vaccines (e.g., malaria, influenza, etc.), as the broad societal discourses around the ‘novelty’ and ‘speed of development’ of the COVID-19 vaccine reflect the unique priority and context of its development (Lurie et al 2020; Troiano & Nardi 2021).”

Second, as R2 rightfully mentions, the scale of media discussion, the ‘infodemic,’ misinformation, disinformation, and the like is significant with respect to this contagion.

Regarding the role of the news media, we believe that it is a crucial strand of research to explain vaccine hesitancy. In fact, Google Scholar provides more than 6,600 results for a search containing “COVID-19” and “Misinformation” and “Disinformation” published in 2020 or 2021. To this very valid suggestion, in the introduction (page 2), we refer to effective communication in the following:

“Other research has sought to appraise the effectiveness of educational and communicative strategies at improving vaccine knowledge and increasing willingness to vaccinate amongst populations (e.g., Gallè et al., 2021; Jin et al., 2021; Motta et al., 2021).”

In the literature review (page 3), we refer to “social media (mis/dis)information” as one of the factors affecting citizens’ decision-making process, and refer to the work of Puri et al. (2020) and Jennings et al. (2021). Although analyzing the role of mass media information and infodemia is out of the scope of our research as our contribution is about the role of timing on vaccine behaviour, we recognize that it is important in the scientific efforts to combat COVID-19. In the new discussion section (line 587), we now include the following and add new references on the topic:

“(…) In any event, deployment of effective communication strategies (e.g., Gallè et al. 2021) that help to provide targeted information that addresses these questions may be useful. This is particularly important given the negative impact of (dis/mis)information on vaccine behaviour around the world (Demuvakor et al. 2021; Dib et al. 2021; Madraki et al. 2021). As put forward by Butcher (2021: 8), scientific communication must take into account readers’ and viewers’ concerns, should be simple and accessible, and should ideally also be capable of performing well in a social media environment.”

New references:

Jennings, W., Stoker, G., Bunting, H., Valgarðsson, V. O., Gaskell, J., Devine, D., ... & Mills, M. C. (2021). Lack of Trust, Conspiracy Beliefs, and Social Media Use Predict COVID-19 Vaccine Hesitancy. Vaccines9(6), 593.

Butcher, P. (2021). Covid-19 as a turning point in the fight against disinformation. Nature Electronics4(1), 7-9.

Demuyakor, J., Nyatuame, I. N., & Obiri, S. (2021). Unmasking COVID-19 Vaccine “Infodemic” in the Social Media. Online Journal of Communication and Media Technologies11(4), e202119.

Dib, F., Mayaud, P., Chauvin, P., & Launay, O. (2021). Online mis/disinformation and vaccine hesitancy in the era of COVID-19: Why we need an eHealth literacy revolution. Human vaccines & immunotherapeutics, 1-3.

Lurie, N., Saville, M., Hatchett, R., & Halton, J. (2020). Developing Covid-19 vaccines at pandemic speed. New England Journal of Medicine382(21), 1969-1973.

Madraki, G., Grasso, I., M. Otala, J., Liu, Y., & Matthews, J. (2021, April). Characterizing and comparing COVID-19 misinformation across languages, countries and platforms. In Companion Proceedings of the Web Conference 2021 (pp. 213-223).

Troiano, G., & Nardi, A. (2021). Vaccine hesitancy in the era of COVID-19. Public Health.

Comment 3: “Moreover, not sure whether this approach may help in the administration decision as does not consider any geographical distribution likely resulting in an inefficient vaccine distribution hence herd immunity if targeted only to willing people that may concentrate more in a specific area than another.”

This is a good point that is particularly relevant in Canada, a vast country known for its regional variations on several outcomes. For that reason, we made sure to include respondents’ province of residence in a robustness check. See page 8:

“(…) but we should note that controlling for age, gender, education, province and francophone does not alter our findings (see Figure SM.1 and Table SM.1 of the Supplementary Material).”

However, we did not provide much context about the relevance of this test and, thanks to R2’s comment, we now do so in the new version of the manuscript (line 350-356):

“Controlling for regions is particularly reassuring as Canada is a vast country with quite a lot of regional variation on different outcomes. Getting the vaccine is the most important preventive measures, but scholars have examined whether there were some regional differences in several preventive measures (including social distancing, wearing a mask, etc.) across the country. Quite surprisingly, Daoust et al. (2021) used 23 survey-wave including more than 22,000 respondents over twelve months and found that differences, when any, were modest.

While the question of how countries should administer their vaccines relative to geographical distribution is beyond the scope of this project, we have added clarification in the conclusion about one of the possible insights of this research: that some of those seen as ‘anti-vax’ now may, actually, still be reachable, if they held these ‘delay’ attitudes:

Indeed, over time, this study suggests there may even be possibilities to convert those who would have described themselves as unwilling to receive a vaccine in a more proximate scenario. 

Comment 4: “This paper is very well written, nevertheless in this form it’s quite long winded, I would suggest to be more concise and straight to the point.”

We thank R2 for this comment. We have, to the best of our ability, tightened up a series of explanations and made better use of the supplemental materials to avoid adding additional length, while trying to balance this with requests for additional explanations by other reviewers. We have also made better use of subheadings to help guide the reader through the different analytical sections, and to break them up into more manageable portions. We address these issues in greater detail with respect to R3’s general comment (#8, below) about the length and flow.

Comment 5: “Finally, I would really appreciate they discuss more the limitations of the study, which would highlight a strong analytical sense and scientific responsibility, meaning that they got the weaknesses of the study and its design from different point of view.”

We thank R2 for this point, which is very fair. We have now revised the discussion section where we discuss the limits of our research. On page 14-15 we now have a more substantive discussion of the limitations, including the specific limitations discussed above. We have also, in a complementary way, tried to reduce instances where there is a risk of ‘overclaiming’ the results (e.g., references to which contexts the study is applicable in). This also adds up to other considerations previously included in the method section (e.g., lines 265-275).

Reviewer 3 Report

Although the topic is interesting, there are some problems in the manuscript.

1. The major issue is the presentations are unclear to the readers. Specifically, the authors mentioned that they have open-ended questions; however, the what the open-ended questions are is described scatter in the Results section. This make the readers hard to have a full picture regarding the study design. Also, the authors have some quantitative data with the use of some closed-ended questions. However, the presentations in the manuscript are also hard for readers to know what exactly the questions look like. 

2. The authors said that they have used some questions to assess the participants' demographics; however, it is unclear what demographics were assessed.

3. Apparently, the authors did not assess important factors related to vaccine hesitancy, such as risk perception, social norms, knowledge. Although this paper aims to use some qualitative features to probe unknown reasons of the vaccine hesitancy, the authors still need to base on some well-established models to assess relevant factors. After all, the evidence has shown, and the authors cannot ignore the evidence.

4. How the chi-square tests and multinominal logistic regressions were done are unclear. The authors did not explicitly describe how they test the chi-square tests and how the logistic regression models were constructed. Therefore, the Results on this part are hard to be interpreted. Moreover, the authors did not control important confounders mentioned above in the logistic regression models (due to the fact that they did not assess). The authors presented coefficients in the logistic regressions, and this is another interpretation problem. Specifically, it is hard to know the meaning of the coefficients when they are log-transformed. The authors should provide odds ratios for better interpretation.

5. Figure 1 is also hard to interpret, I cannot understand what the meaning is for the y axis.

6. Although I am not an expert in qualitative study, my feelings on the qualitative analysis and the qualitative results in the present manuscript are they are not done in an academic and standardized way. Maybe I am wrong; however, from my past readings in qualitative studies, I never read anything look like the presentations in the manuscript.

7. It is unclear how the authors make the random assignment for the participants to answer the quantitative questions.

8. Overall, the presentations throughout the manuscript are lengthy and hard to follow. The logical flows are not smooth and I cannot link the literature reviewed in the Introduction with the present study design. 

Author Response

Reviewer 3 raised eight comments to be addressed:

Comment 1: “The major issue is the presentations are unclear to the readers. Specifically, the authors mentioned that they have open-ended questions; however, the what the open-ended questions are is described scatter in the Results section. This makes the readers hard to have a full picture regarding the study design. Also, the authors have some quantitative data with the use of some closed-ended questions. However, the presentations in the manuscript are also hard for readers to know what exactly the questions look like.”

We thank Reviewer 3 for this helpful comment, which encouraged us to make sure the design is clear from the beginning. We have ensured that the open-ended questions are explained in the methods section (p. 5) so they don’t appear by surprise in the results. We also provide screenshots in the supplemental materials to ensure readers see exactly what each treatment condition looked like.

Comment 2: “The authors said that they have used some questions to assess the participants' demographics; however, it is unclear what demographics were assessed.”

We realize that we were not clear in the previous iteration of our manuscript and thank R3 for pointing this out. The sociodemographic variables were used during the data collection process, as well as for a robustness check. We now clarify it in the new version of the manuscript:

“Once quota was met, no further respondents were recruited within those demographic criteria. Overall, the sample closely mimics the Census’ data on variables like age, gender, provinces (see Table SM.2 in the Supplementary Material for more detailed comparison). The survey also collected responses on a variety of other issues, including demographic variables which were investigated as possible cofounders in a further robustness check later in the results section.”

Given both this reviewer’s and R2’s comment about length, we have placed Table SM.2 in the supplementary material, but do provide it to help readers better verify the demographics of our respondents:

Table SM.2. Comparison between survey sample and Canadian census

2016 Canadian census

Survey sample

Gender

Women

50.9%

52%

Men

49.1%

48%

Age

18-24

10.9%

10.2%

25-34

16.4%

15%

35-44

16.2%

16.7%

45-54

17.9%

20.2%

55-64

17.5%

17.5%

65+

21.1%

20.5%

Education

Postsecondary certificate, diploma or degree

55.3%

48.6%

Province

Ontario

38.3%

34.4%

Quebec

23.2%

20.9%

British Columbia

13.2%

12%

Alberta

11.6%

10.7%

Manitoba

3.6%

5.2%

Saskatchewan

3.1%

4.7%

Eastern provinces

6.6%

12.2%

Comment 3: “Apparently, the authors did not assess important factors related to vaccine hesitancy, such as risk perception, social norms, knowledge. Although this paper aims to use some qualitative features to probe unknown reasons of the vaccine hesitancy, the authors still need to base on some well-established models to assess relevant factors. After all, the evidence has shown, and the authors cannot ignore the evidence.”

We are thankful to R3 for this comment and agree wholeheartedly with the importance of considering factors involved in vaccine hesitancy like risk perception, social norms, and knowledge. These are important points, and we have attempted to touch on them on within the review of critical factors in section 2 of the manuscript (among others, see line 127-133).

Given the feedback by both R2 and R3 about the length and complexity of the article already, we felt/feel as though it is important to keep our focus on the primary contribution: isolating and investigating the role of timing as a contributing factor to hesitancy. As we now note in the discussion (added in lines 811-815), future work exploring the intersections and interactions of these different factors will be very valuable.

There is, however, another possible concern: that risk perception could explain the variation we saw. To address this concern, our experiment used randomized assignment of participants into each treatment to avoid any systematic bias. We now explain this more clearly on page 5:

“Participants were randomly assigned to each of these four experimental variations using the randomization within the survey platform (Qualia Analytics). Each respondent had a probability of 0.25 to receive every four treatments. The randomisation is key as it allows us to be very confident that all potential cofounders are equally distributed across the different groups.”

That said, we understand R3’s worry given that other factors that they mention can be very important. Fortunately, we did collect data on issues of risk perception, social norms, and knowledge as part of the broader survey project, and analyze it in more detail elsewhere (e.g., Kennedy et al. 2020; Nelson et al., under review). As an example, we can use two questions related to risk perceptions to explore the issues raised by R3. Both questions used an agree-disagree scale. The first item was “I would worry about exposing the people I live with to the virus.” The second item was “new vaccines carry more risks than older vaccines.”

If randomisation of participants to each treatment condition was successful, we should see a very similar distribution for these variables across the different treatment groups. This is exactly what we observe in Figures SM.3. and SM.4. below. The distribution are strikingly similar.

Figure SM.3. Worry about risks for family.

Figure SM.4. Risk perceptions of vaccines

Thanks to this helpful comment – and in full agreement of the importance of these factors – we now include these figures in the supplementary material as an additional check (again, given both reviewer’s concerns about the length and complexity of the text already), and highlight their presence in the main text on page 9 for interested readers.

Comment 4: “How the chi-square tests and multinominal logistic regressions were done are unclear. The authors did not explicitly describe how they test the chi-square tests and how the logistic regression models were constructed. Therefore, the Results on this part are hard to be interpreted. Moreover, the authors did not control important confounders mentioned above in the logistic regression models (due to the fact that they did not assess). The authors presented coefficients in the logistic regressions, and this is another interpretation problem. Specifically, it is hard to know the meaning of the coefficients when they are log-transformed. The authors should provide odds ratios for better interpretation.”

These are fair and helpful points. First, on the Chi-square: we now clarify that we conducted a Pearson’s chi-square, and that we did so using the Stata option when cross-tabulating the outcome over the treatment groups. The degrees of freedom (9), number of observations (2,658), the chi-square value (27.466) and the p-value (0.001) are also all mentioned. See the lines 315-318:

“In a first step, we conducted a Pearson’s Chi-square test (using Stata chi2 option) when cross-tabulating vaccine intention across the four treatments. As expected, we find a different that is statistically significant, whereas X2(9, N=2,568)=27.466, p=0.001”

Second, we agree that using log-odds coefficients is not helpful for the interpretation. Using odds ratio helps, but it is still quite complicated since both the IV and the DV are categorical and interpreted in light of a reference category. This is why we focused on plotting the predicted probabilities for each outcome of the DV based on each treatment group (the IV). We realize that we were not clear about this (see also the next comment). We made the following revisions to the manuscript: we now explain in the text that we focus on predicted probabilities (see line 320); we now make it clear in Figure 1 that the y-axis corresponds to predicted probabilities; we now show, as suggested by R3, the odds ratio in the regression table.

Third, on the confounders: while we believe that it is important to acknowledge the importance of several variables (as we believe we do in section 2 of the manuscript), the experimental design allows us to naturally control for these factors. For more discussion, see our response to the previous comment and the example regarding risk perceptions.

Comment 5: “Figure 1 is also hard to interpret, I cannot understand what the meaning is for the y axis.”

As mentioned in our response above, we realize that we were not clear and thank R3 for the comment. It corresponds to the predicted probabilities. We revised the Figure and now it now clear.

Comment 6: “Although I am not an expert in qualitative study, my feelings on the qualitative analysis and the qualitative results in the present manuscript are they are not done in an academic and standardized way. Maybe I am wrong; however, from my past readings in qualitative studies, I never read anything look like the presentations in the manuscript.”

We thank R3 for this comment, which is close to the heart of authors 1, 3, and 4, who are all qualitative researchers by training. Because of the nature of this mixed methods investigation – and particularly the length issues mentioned by both R2 and R3 – it is not possible for us to conduct a full-scale qualitative analysis. Rather, the aim is to provide a brief overview of the content and themes participants shared, such that it can support the further development of more fully fledged qualitative analyses. More importantly, we believe that there is value to the integrated whole of this mixed methods approach: looking at both the quantitative survey experiment and the qualitative responses of our participants in an integrated way offers a broad picture that cannot be captured in an approach focused exclusively on either the quantitative experiment or the qualitative responses.

This comment was very helpful, however, in helping us to restructure the presentation of the qualitative results to make them easier to follow. To improve clarity, we separated the inductive coding and close reading portions with sub-headers. We have also reviewed and edited these portions to make them easier to read. Finally, we discussed the limitations and possibilities for future research in the discussion.

Comment 7: “It is unclear how the authors make the random assignment for the participants to answer the quantitative questions.”

Comment 8: “Overall, the presentations throughout the manuscript are lengthy and hard to follow. The logical flows are not smooth and I cannot link the literature reviewed in the Introduction with the present study design.”

We thank R3 for this comment, and have carefully reviewed the manuscript to try to make its flow clearer, more concise, and easier for the reader. We have done this in a few ways:

  • We have separated out the literature review section into its own section, which should help to make it easier for readers to find our review of issues like misinformation and vaccine hesitancy. We have also structured the literature review section to progress from general factors towards previous investigations exploring delay/wait phenomenon, so that it’s clearer how the literature review connects to our work.
  • We have separated out subsections within the analysis section to make it easier to follow, particularly with respect to the qualitative analysis. We have also added more transitions to help the reader progress from section to section.
  • We have made increased use of the supplemental materials to help ensure content is available (e.g., additional statistical analysis, examples of the user experience) without cluttering the core text.

Round 2

Reviewer 2 Report

In this version, and in the reply letter, authors argumentated and sastisfied all the concerns I raised from the first version.

Moreover, I am glad they caught my suggestion discussing more deeply the limitations and weaknesses of their study hence strengthening the scientific sound.

FInally, although I still found the narrative quite long the MS in this form is more convincing and certainly represents a valide contribute on the topic.

Reviewer 3 Report

Although in my first reading of the manuscript I do not think that the authors can address my concerns successfully, the authors did complete a fantastic job to address all my concerns. They have remediated their manuscript in a satisfactory way. I have no more comments and am happy with the present revision. Good job. 

This manuscript is a resubmission of an earlier submission. The following is a list of the peer review reports and author responses from that submission.

Round 1

Reviewer 1 Report

The investigators performed a mixed method analysis using cross-sectional data among 2,602 online survey participants in Canada. The objective of this study was to examine the timelines of COVID-19 vaccine availability associated with COVID-19 vaccine intentions. This study is important regarding the public health significance of the COVID-19 pandemic. The investigators have a well written introduction/background. However, there are some major considerations necessary for this investigation to become slightly more coherent and to effectively present the investigators’ work. My thoughts and suggestions are below.

Major Comments

1) Within the introduction, there are a couple of places where interpretations of results i.e., discussion is present. On page 2 lines 72 through 79 and lines 87 through 92, this text can be reframed/transitioned to the discussion (possibly first paragraph). This text is describing the findings of the current study, and it is not commonplace to describe the findings during the introduction and background – before introduction of research hypotheses and objectives.

2) The most concerning feature of this study is the structure of the study design, specifically the implementation of the survey questions and corresponding explanatory variable (in this case, described as experimental variations). The investigators distributed a survey among nearly 3,000 online participants but randomly assigned participants to receiving the following four questions “If a coronavirus vaccine was available to you”: (1) today, (2) in one month, (3) in six months, or (4) in one year “would you get vaccinated, or not?” There are several concerns to be considered including information biases, but most importantly why limit certain questions to randomly selected participants? What is the rationale? Further, these ‘experimental variations’ create subsamples (i.e., participants within a larger study with data collected on select variables of interest). Instead, why weren’t all participants asked the likelihood of getting a coronavirus vaccine at the varying time points, followed by the other possible survey questions on page 4 lines 192 – 197? The current methodology makes the interpretations difficult to follow.

3) The investigators should consider providing more detail in the methods on variables, how they were categorized, design, consenting or IRB, and the study population. There are several guides that may assist in the strengthening of the current methods, analysis, and results.

I recommend using either:

(a) Strengthening the reporting of observational studies in epidemiology (STROBE) to have a more transparent manuscript: https://www.strobe-statement.org/index.php?id=available-checklists.

(b) Kelley K, Clark B, Brown V, Sitzia J. Good practice in the conduct and reporting of survey research. Int J Qual Health Care. 2003;15(3):261-266. Located here:

3) Another major consideration is the presentation of methods and results. The investigators need to provide more information regarding analysis, if any (which there is because supplemental describes logistic regression), level of significance, specific tests used, etc. What type of software were used to perform these analyses? It is important to describe these statistical methods for the reproducibility of research.

Minor Comments

1) Again, the introduction is very nicely written. However, on page 2 lines 56 through 63, it would be helpful to qualify that “vaccines” are “COVID-19 vaccines” during this text and discussion.

2) Also, within the introduction could the investigators explain more how trust with certain populations may play a significant role (i.e., page 3 lines 107-108)?

3) It is very unorthodox to use language such as “As clear from eyeballing Figure 1” as given on page 5 line 240. This is inappropriate for results description and text. This language may be resultant from lack of utilizing hypotheses testing and corresponding p values and/or parameter estimates that describe the differences/comparisons between groups. For example, the text “This percentage increases to 47% (“In one month”), 54% (“In six months”), and 58% (“In one year”)” could all be tested using common statistical approaches, i.e., Chi-square tests for comparisons of proportions between groups.

4) It may be helpful to provide descriptive table 1 for the quantitative portion of the study. Readers/fellow scientists have no idea of sample sizes per group, measures of central tendency and dispersion for any of the presented data.

5) How was missing handled for the quantitative portion of the analysis?

Author Response

Thank you very much for your detailed and most helpful review. Please see the attached file for our responses.

Reviewer 2 Report

First of all, I am grateful for the opportunity to review this paper. COVID-19 is an ongoing pandemic that has resulted in global health, economic and social crises. Actually, the vaccination campaign is the first method to counteract the COVID-19 pandemic; however, sufficient vaccination coverage is conditioned by the people’s acceptance of these vaccines. In this context, the paper under review is aimed at investigating the impact of different timing scenarios and exploring the qualitative explanations provided about these decisions.

The article is interesting and may provide important information for public health, but it must be improved to be suitable for publication in an international journal.

Introduction: The authors should make it clear about what is the gap in the literature that is filled with this study? First of all, the general acceptance of the COVID-19 vaccine and impact of communication campaign and media communication must be better discussed (refer to Gallè, F. et al Knowledge and Acceptance of COVID-19 Vaccination among Undergraduate Students from Central and Southern Italy. Vaccines 2021, 9, 638). What is the contribution of the study to the literature? What are the implications of the study?

Methods: A questionnaire was used for the first part of the study, but how was the questionnaire validated (face validity, intelligibility, reliability)? Moreover, it is not clear how the Authors selected the enrolled people. Is the sample representative of what population? What is the minimum sample? What is the power of the study?   

Ethical Issue: although an anonymous questionnaire is used, an ethical approval is necessary. An ethical committee should approve the study protocol, and a reference number should be reported.

Discussion: I also suggest expanding. Emphasize the contribution of the study to the literature, the implications and recommendations based on previous experience also in other population groups also discussing effectiveness of the information strategy (refer to Gallè, F. et al Knowledge and Acceptance of COVID-19 Vaccination among Undergraduate Students from Central and Southern Italy. Vaccines 2021, 9, 638). Limits section must be better described.

Author Response

Thank you for your detailed and most helpful review. Please see the attached file for our responses.

Round 2

Reviewer 1 Report

I am not able to provide a review as the investigators only responded to Reviewer 2. There is no response to Review 1 in the letter.

Reviewer 2 Report

the paper has been improved and, in my opinion, it is now suitable for publication

Round 3

Reviewer 1 Report

The investigators performed a mixed method analysis using cross-sectional data among 2,602 online survey participants in Canada. The objective of this study was to examine the timelines of COVID-19 vaccine availability associated with COVID-19 vaccine intentions. This study is important regarding the public health significance of the COVID-19 pandemic. The investigators have a well written introduction/background. The investigators attempted to assuage my concerns, however, this manuscript in its current form is not of good research methodology, design, and reporting of methods and results. Below are my concerns.

(1) Instead of presenting manuscript amendments as footnotes, these updates and many of the currently presented footnotes would serve better if implemented within the methods text. Moreover, add this language to the text of the methods or corresponding sections.

(2) The investigators should consider providing a statistical analyses section.

(3) I strongly believe that the implementation of the survey methodology and questioning of primary outcome(s) “intentions to receiving COVID-19 vaccine” are confusing/flawed and are not clearly presented for interpretation of these data. It is still unclear how the following question is being examined: “whether the timing of vaccine rollout has significant impact on public decision making.”

(4) There are many typographical and grammatical errors throughout the manuscript that must be corrected before this manuscript is considered for publication.